

# Scaling the effects of ocean acidification on coral growth and coral–coral competition on coral community recovery

Nicolas R. Evensen[1,2,3], Yves-Marie Bozec[3], Peter J. Edmunds[2] and Peter J. Mumby[3]

[1] Department of Biological Sciences, Old Dominion University, Norfolk, VA, United States
[2] Marine Spatial Ecology Lab, ARC Centre of Excellence for Coral Reef Studies and School of Biological Sciences, University of Queensland, St. Lucia, QLD, Australia
[3] Department of Biology, California State University, Northridge, Northridge, CA, United States

Corresponding author
Nicolas R. Evensen,
nicolas.r.evensen@gmail.com

## ABSTRACT

Ocean acidification (OA) is negatively affecting calcification in a wide variety of marine organisms. These effects are acute for many tropical scleractinian corals under short-term experimental conditions, but it is unclear how these effects interact with ecological processes, such as competition for space, to impact coral communities over multiple years. This study sought to test the use of individual-based models (IBMs) as a tool to scale up the effects of OA recorded in short-term studies to community-scale impacts, combining data from field surveys and mesocosm experiments to parameterize an IBM of coral community recovery on the fore reef of Moorea, French Polynesia. Focusing on the dominant coral genera from the fore reef, *Pocillopora*, *Acropora*, *Montipora* and *Porites*, model efficacy first was evaluated through the comparison of simulated and empirical dynamics from 2010–2016, when the reef was recovering from sequential acute disturbances (a crown-of-thorns seastar outbreak followed by a cyclone) that reduced coral cover to ~0% by 2010. The model then was used to evaluate how the effects of OA (1,100–1,200 µatm $pCO_2$) on coral growth and competition among corals affected recovery rates (as assessed by changes in % cover $y^{-1}$) of each coral population between 2010–2016. The model indicated that recovery rates for the fore reef community was halved by OA over 7 years, with cover increasing at 11% $y^{-1}$ under ambient conditions and 4.8% $y^{-1}$ under OA conditions. However, when OA was implemented to affect coral growth and not competition among corals, coral community recovery increased to 7.2% $y^{-1}$, highlighting mechanisms other than growth suppression (i.e., competition), through which OA can impact recovery. Our study reveals the potential for IBMs to assess the impacts of OA on coral communities at temporal and spatial scales beyond the capabilities of experimental studies, but this potential will not be realized unless empirical analyses address a wider variety of response variables representing ecological, physiological and functional domains.

## INTRODUCTION

Evaluation of the effects of ocean acidification (OA) on marine organisms (*Kroeker et al., 2010*; *Wittmann & Pörtner, 2013*) supports the general conclusion that biogenic calcification will be depressed by rising seawater $pCO_2$ (*Chan & Connolly, 2013*; *Kornder, Riegl & Figueiredo, 2018*; *Kroeker et al., 2013*). For scleractinian corals, elevating $pCO_2$ above current atmospheric levels (i.e., ~400 µatm) mostly results in decreased calcification (*Chan & Connolly, 2013*; *Comeau et al., 2014a*), although species differ in the sensitivity of net calcification to high $pCO_2$ (*Comeau et al., 2014b*; *Edmunds, Brown & Moriarty, 2012*). Additionally, OA can reduce coral recruitment (*Albright & Langdon, 2011*; *Doropoulos et al., 2012*; *Fabricius et al., 2017*), although it is thought to have little effect on the survival of juvenile or adult corals (*Kroeker et al., 2013*).

Beyond affecting vital rates (like recruitment), elevated $pCO_2$ also has the potential to alter the synecology of coral reefs, for example, by altering competitive dynamics between corals and algae (*Diaz-Pulido et al., 2011*), as well as among corals (*Evensen & Edmunds, 2016*; *Horwitz, Hoogenboom & Fine, 2017*). It is challenging, however, to predict how the effects of OA on ecological and demographic processes will scale up to affect coral community dynamics (*Albright & Langdon, 2011*; *Edmunds et al., 2016a*). This challenge is intensified as the prediction extends further in to the future, because predictions over decades or centuries requires the effects of OA to be considered against a backdrop of chronic and acute disturbances, such as major storms and ocean warming (*Hughes et al., 2017*). The most ambitious predictions also require consideration of the potential for acclimatization or adaptation (*Schoepf et al., 2017*; *Kurihara et al., 2020*).

Field and mesocosm studies of the response of coral communities to OA face significant challenges of sustaining experiments for ecologically relevant durations (i.e., months to a year), and procuring sufficient replicate organisms within the constraints of permitting and ethics. Modelling provides a tractable alternative to empirical studies, with individual-based models (IBMs) supporting accurate simulations of the spatial dynamics of coral colonies within reef communities (*Mumby, Hastings & Edwards, 2007*; *Sandin & McNamara, 2012*). IBMs can also support decadal-scale projections of coral population structure when they include empirical determinations of vital rates (e.g., recruitment and growth), and mechanistic understanding of ecological processes, such as competitive interactions. Despite the potential of IBMs to evaluate the response of coral communities to contrasting disturbance regimes (*Muko et al., 2014*; *Ortiz et al., 2014*), they have not been widely applied to study the effects of OA on coral reefs. To date, the most comprehensive modelling approach forecasting the impacts of OA on corals included one coral (*Acropora*) and one macroalgal (*Lobophora*) genus, to demonstrate that increased intensity of competition between corals and algae can interact with OA to lower coral calcification (*Anthony et al., 2011*). In complex coral reef communities including a greater number of coral and algal taxa, taxonomically variable responses to OA, and a wider diversity of perturbed ecological interactions, are likely to result in the detection of strong, community-wide effects. Consideration of more complex ecological situations will be

necessary to improve the ecological relevance and accuracy of models seeking to determine how reef communities will change upon long-term exposure to OA.

The present study sought to expand on the use of modelling approaches to scale up the impacts of elevated $pCO_2$ across larger spatio-temporal scales, using the IBM approach of *Mumby, Hastings & Edwards (2007)* to evaluate how coral community dynamics will change in a future strongly affected by OA. In this application, the effects of high $pCO_2$ on coral growth were species specific, and coral–coral competition was implemented to modulate community dynamics in a manner that differed among $pCO_2$ regimes. The model was implemented for the coral community on the fore reef of Moorea, French Polynesia (Fig. S1), and it utilized two domains of empirical data. First, results from manipulative experiments were utilized to quantify the effects of elevated $pCO_2$ on the growth (linear extension) and capacity for competition among the four common coral genera (*Pocillopora*, *Acropora*, *Montipora* and *Porites*) found on the fore reef (10-m depth) of Moorea (*Evensen, Edmunds & Sakai, 2015*; *Evensen & Edmunds, 2016*). Second, vital rates for each genus were obtained from an ecological time-series of coral community structure at 10-m depth on fore reef of Moorea (*Edmunds, 2018a*; *Holbrook et al., 2018*). This time series extended from 2005 to present, but only results from 2010–2016 were employed in the present analysis with the rationale that this period spans a remarkable time in the history of these reefs, commencing with virtually no coral as a result of COTs and a cyclone (*Kayal et al., 2012*), and ending with 66% coral cover in 2016 (*Edmunds, 2018a*). The model was used to determine how the effects of OA on coral growth and competition affect coral populations and the rate of coral community recovery following acute disturbances.

## METHODS

### Model overview

An individual-based, spatially explicit model of coral population structure, developed by Mumby and colleagues (*Bozec et al., 2016*; *Mumby, Hastings & Edwards, 2007*; *Ortiz et al., 2014*), was used to project coral community cover in Moorea. The model was developed for Caribbean reefs (*Mumby, Hastings & Edwards, 2007*) and later extended to Indo-Pacific reefs (*Ortiz et al., 2014*). Here, the model is adapted to the fore reef of Moorea by parameterization with empirical data from this location that describe vital rates for the common coral genera, as well as the outcomes on colony growth of coral–coral competition under OA (*Evensen & Edmunds, 2016*). Permits for fieldwork were issued by the Haut-commissariat de la République en Polynésie Française (DRRT) (Protocole d'Accueil 2010–2011, 2011–2012, 2012–2013, 2013–2014, 2014–2015, 2015–2016 to PJE).

The model simulated the fate of coral colonies on a square lattice of 400 cells, with each cell representing 1 m$^2$ of reef substratum. Cells in the model could be occupied by multiple coral colonies of different genera, with each colony quantified by its cross-sectional (circular) planar area. The model advanced in 6-month increments, with colony size (in cm$^2$) and density updated every time step. Coral colonies increased in size based on: (1) genus-specific rates of linear (radial) growth (Table S1), (2) competition with con- and hetero-specifics (Table 1), and (3) the sensitivity of coral growth (i.e., linear

**Table 1 Parameterization used to alter coral growth rates as a function of contact with conspecific and heterospecific coral competitors.** Under ambient conditions, change in growth as a function of contact with conspecifics is determined by a linear relationship (Equation 1, below) and change in growth as a function of contact with heterospecifics ($b_h$) is determined by an exponential decay function (Equation 2, below). Under OA, change in growth as a function of contact with conspecifics ($b_c$) and heterospecifics ($b_h$) are determined by an exponential decay function (Equation 2). Value are unitless as they represent proportional changes in lateral extension as the percent contact increases, compared to maximum growth reported in Table S1 (scaled from 0–1).

| Equation 1 | $= 1 - a \times percent\ coral\ contact$ | | | |
|---|---|---|---|---|
| Equation 2 | $= exp \times (-b \times percent\ coral\ contact)$ | | | |
| **RCP scenario** | *Acropora* | *Montipora* | *Pocillopora* | *Porites* |
| Ambient | a = 0.005 | a = 0.005 | a = 0.005 | a = 0.005 |
| | $b_h$ = 0.01 | $b_h$ = 0.022 | $b_h$ = 0.018 | $b_h$ = 0.026 |
| OA (RCP8.5) | $b_c$ = 0.0097 | $b_c$ = 0.0097 | $b_c$ = 0.0097 | $b_c$ = 0.0097 |
| | $b_h$ = 0.035 | $b_h$ = 0.077 | $b_h$ = 0.063 | $b_h$ = 0.091 |

extension) to elevated $pCO_2$ (Table S2). The effects of competition and OA on colony growth rates are detailed below.

## Modelling competition between coral colonies

Ecological realism was added to the model by (1) addressing the competitive interactions that occur when coral colonies encounter one another while growing and occupying space on the benthos, and (2) evaluating how these interactions might be modified by future environmental conditions. To capture the effects of coral–coral competition, we first determined the extent to which the perimeter of coral colonies encounter (i.e., touch) one another as they spread across the benthos, growing symmetrically as planar circles. This information was extracted from 0.25 m$^2$ photoquadrats recorded annually (n = ~40 y$^{-1}$) at fixed positions along a transect at 10-m depth at the LTER1 site on the north shore of Moorea. The photoquadrats used in the present study came from 2010–2016, which is the period of recovery of the coral community following the outbreak of COTs and a cyclone; these data were proportionately scaled to 1 m$^2$ to match the size of the model cells. In each photoquadrat, the distance among colony centroids and the radius of each colony following 6 months of growth (i.e., the model time step) were measured and used to calculate the length of the perimeter of each colony in contact with another (Figs. 1A–1B). For these calculations, colonies of all four genera were assumed to extend linearly in planar view by 1.6 cm every 6 months, which corresponds to the average linear growth rate of all four genera over a 6-month period (Table S1). Full code detailing the calculations can be found in Supplemental Material.

Data recorded from the photoquadrats were used to parameterise a model simulating random scenarios of placement of coral colonies varying in size and density within model cells (Fig. 1C). Colony sizes were generated at random following the log-normal distribution of colony sizes observed in the LTER1 photoquadrats from 2010–2016, with colony densities ranging from 2 to 35 colonies m$^{-2}$ to match the observed range of colony densities recorded between 2010–2016. This produced a relationship of the rate at which coral colonies engage in competitive encounters with neighbouring colonies (i.e., they

a) Data extraction from photoquadrats        b) Calculating contact between corals

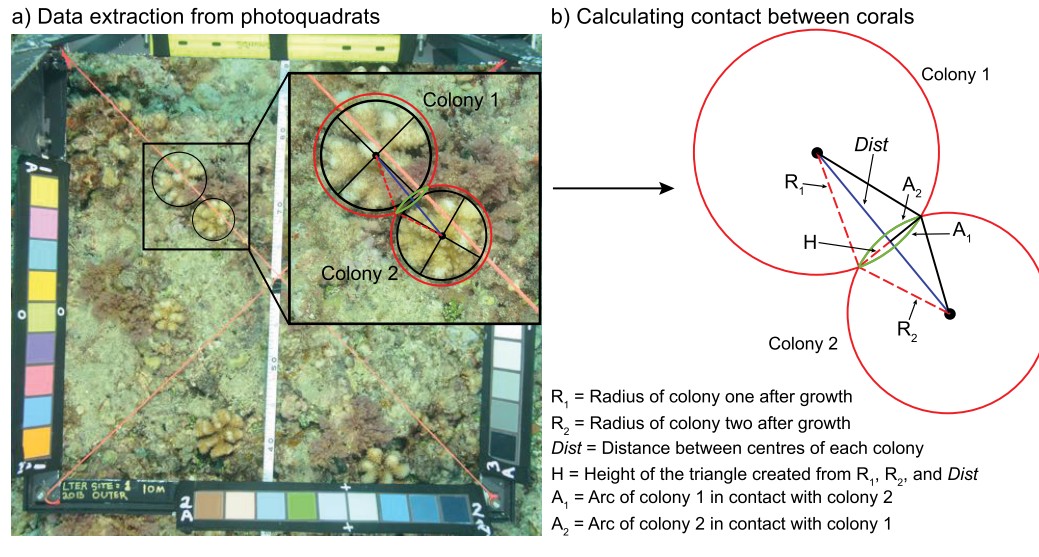

R₁ = Radius of colony one after growth
R₂ = Radius of colony two after growth
*Dist* = Distance between centres of each colony
H = Height of the triangle created from R₁, R₂, and *Dist*
A₁ = Arc of colony 1 in contact with colony 2
A₂ = Arc of colony 2 in contact with colony 1

c) Average contact between colonies as a function of coral cover

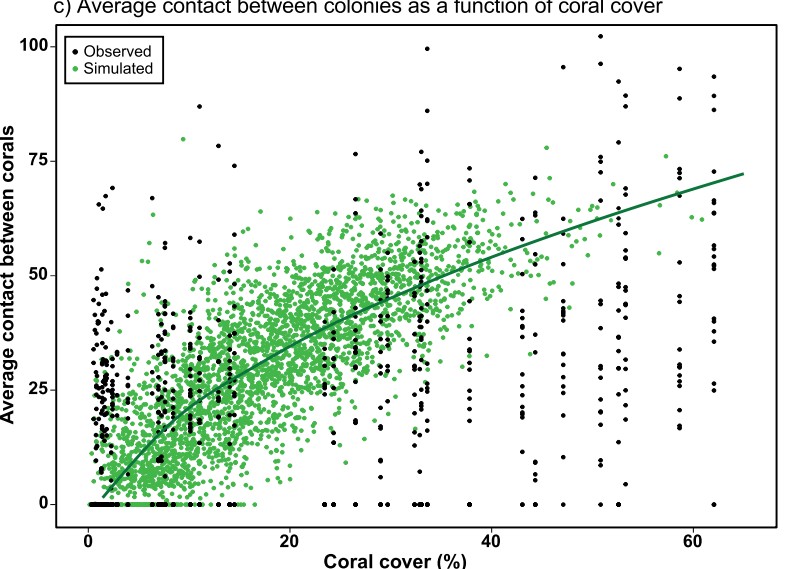

**Figure 1 Measuring coral-coral competition in photoquadrats.** (A) Measurements of the diameters and circumferences of two *Pocillopora* colonies (black lines and circles) and their anticipated growth over 6 months (i.e., one model time step) (red circles), based on growth rates measured in experiments in Moorea (*Evensen & Edmunds, 2016*) . The blue line represents the distance between colony centres, with red dashed lines representing the anticipated radii of each colony after growth and the green lines representing the section (arc) of each colony in contact with the other colony after growth. (B) Using the distance between colony centres (*Dist*) and anticipated radii of each colony after growth (R₁ and R₂) to calculate the height of the triangle (H), which was then used to calculate the length of the contact arc between corals (A₁ and A₂) using Heron's Formula (*Weisstein, 2021*). (C) Average relationship between coral cover and the average percent contact among colonies (proportion of the perimeter of corals in contact with one or multiple colonies). Black circles represent the observed relationship between coral cover and contact from photoquadrats recorded at LTER1 from 2010–2015 ($n = 65$), while green circles and the darker green line represent the simulated relationship between coral cover and contact based on a random distribution of corals within a cell ($n = 3,400$). Full details of the equations and code used to calculate and simulate coral competition are provided in Supplemental Material.

contact one another) as coral cover increases. Measurements were recorded within the 1 m$^2$ cells of the model, in which the size of colonies (but not their exact location) was recorded at each time step. The relationship between coral cover and mean percentage of the perimeter of each colony that contact adjacent colonies (i.e., extent of competitive encounters among corals) was fitted by a linear model after square-root transformation of coral cover (Fig. 1C). The linear relationship between coral cover and competition among corals was integrated into the IBM by generating an average value of contact between colonies, within the 95% CI of predictions from the linear model, for every grid cell as a function of coral cover.

Following quantification of the rate and extent of contact among coral colonies (i.e., competition) as a function of coral cover, the effect of this contact on linear extension was integrated into the model designed to project coral cover over time. The outcome of coral–coral contact varied as a function of competitor identity (i.e., con- versus hetero-specifics) and the extent of contact (proportion of the colony perimeter) with the competitor. The effect of contact with conspecifics and heterospecifics under ambient and OA conditions were based on results from *Evensen & Edmunds (2016)*, in which the consequences of competitive encounters between *Pocillopora verrucosa* and conspecifics and heterospecifics (*A. hyacinthus*) in Moorea were evaluated by the extent to which planar linear extension of *P. verrucosa* was depressed by these encounters under ambient (411 μatm) and elevated pCO$_2$ (1,033 μatm).

*Evensen & Edmunds (2016)* conducted experiments in which colonies of *P. verrucosa* were surrounded with 2–4 colonies of con- or hetero-specifics in 500-L outdoor flumes (Fig. 2A), with the impact of competition among these colonies inferred from changes in linear growth of *P. verrucosa* after 4 weeks. The dependence of linear extension on percentage contact with competitors under ambient and elevated pCO$_2$ was fitted with either a simple linear model (*Equation 1*, Table 1) or an exponential decay function (*Equation 2*, Table 1), depending on the best model fit, as estimated by Akaike information criterion (AIC).

Other than the aforementioned data for *Pocillopora*, data were unavailable to quantify the growth response of *Acropora*, *Porites* and *Montipora* when competing with conspecifics or heterospecifics, although evidence from the Red Sea suggests growth is affected equally by competition for each of these genera (*Horwitz, Hoogenboom & Fine, 2017*). Thus, the empirical relationship between linear growth and percentage contact with conspecifics determined for *Pocillopora* was assumed to apply to all genera in the present study, although linear extension rates in the absence of competition and the effect of elevated pCO$_2$ remained specific to each genus. In turn, genus-specific responses of corals to heterospecific competition were parameterized by modifying the relationship between linear growth and contact with heterospecifics based on previously reported competitive hierarchies that include the present genera (i.e., *Acropora > Pocillopora > Montipora > Porites*; based on *Dai, 1990*; *Connell et al., 2004*; *Evensen, Edmunds & Sakai, 2015*; *Horwitz, Hoogenboom & Fine, 2017*). As the relationship between linear growth and contact with heterospecifics was best described by an exponential decay model, genus-specific responses of corals to heterospecific competition were implemented by

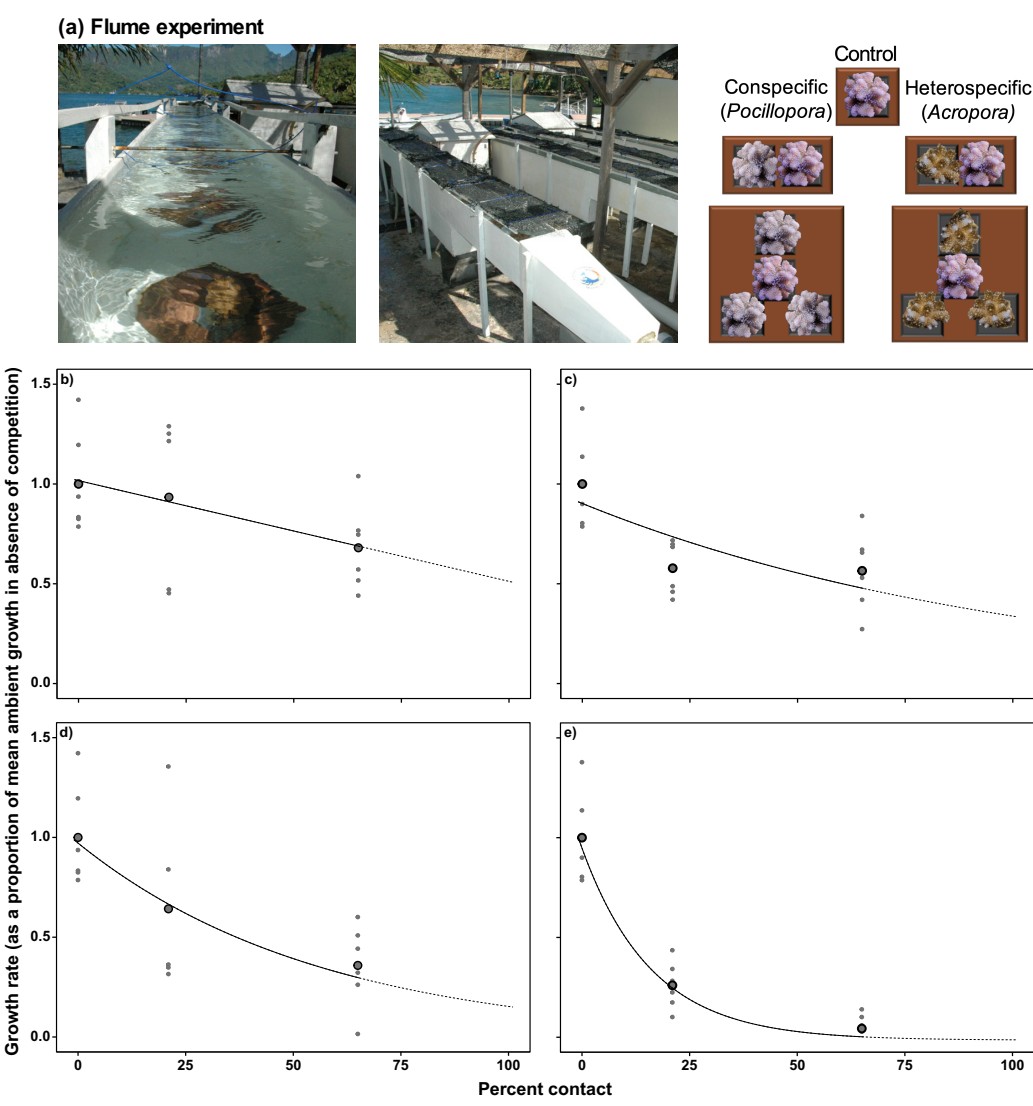

**Figure 2 Determining coral linear growth rates as a function of competition.** (A) Photographs and diagram of the experimental set up from *Evensen & Edmunds (2016)* used to assess linear growth of *Pocillopora verrucosa* over 28 d as a function of contact with surrounding coral competitors, under ambient (~400 µatm) and elevated $pCO_2$ (~1,030 µatm) in 500-L outdoor flumes (each 5.0 × 0.3 × 0.3 m). Relationships between coral colony planar growth (as a proportion of the mean growth rate under ambient $pCO_2$ conditions in the absence of competition) and percent of the perimeter of a colony in contact with conspecifics (B) and heterospecifics (C) under ambient $pCO_2$ conditions (~400 µatm), and contact with conspecifics (D) and heterospecifics (E) under elevated $pCO_2$. Values are based on mean growth rates (larger black dots), with small grey dots representing individual replicates. Relationships within the measured values are represented by the solid black lines, with the extrapolation to 100% contact represented by the dotted line. Equations used to implement the effects of competition on colony growth are provided in Table 1.

altering the rates at which linear growth of *Pocillopora* declined as a function of contact with heterospecifics. Relative to *Pocillopora*, rates of decline in linear growth as a function of increasing contact with heterospecifics were 44% slower for *Acropora*, and 22% and 44% faster for *Montipora* and *Porites*, respectively (Table 1).

## Evaluating model accuracy: simulating coral community recovery after disturbances

The model was implemented with the coral cover of each genus recorded at 10-m depth on the north shore of Moorea in April 2010, which consisted of 0.2% cover of *Porites* following several years of a COTS outbreak and a cyclone in February 2010 (*Edmunds, 2018a*). By 2016, coral cover at the same site had reached 66 ± 3%, with *Acropora, Pocillopora, Porites,* and *Montipora*, contributing >97% of the cover. Our model, therefore, focused on: *Pocillopora*, which in this location includes at least *P. verrucosa, P. meandrina, P. eydouxi, P. effuses*, and several unnamed haplotypes (*Edmunds et al., 2016b*; *Burgess et al., 2021*); *Acropora* spp. which includes at least 15 species (*Bosserelle et al., 2014*); *Montipora* spp. which includes at least nine species (*Bosserelle et al., 2014*); and *Porites* which was evaluated as the functional group "massive *Porites*" including *P. lutea, P. lobata* and *P. australiensis* (*Edmunds, 2018a*).

Conversely, coral-macroalgal competition was not modelled in the present study as macroalgae has remained at <5% cover at 12-m depth on the fore reefs of Moorea since 1991 (*Adjeroud et al., 2018*) and thus is not a major competitor for space at this fore reef site. Indeed, the most abundant macroalgae at this site, *Asparagopsis taxiformis*, did not exceed 5% cover between 2010 and 2016 in the photoquadrats and steadily decreased in cover as coral cover increased. As such, coral-macroalgal interactions were not common during the recovery of the coral community.

Each time step in the model (6 months) was informed with quantitative vital rates characterizing the four genera (described below; Table S1). The accuracy of the population projections was determined by comparing the projected coral cover by year with the empirical data recorded from 2010 to 2016 at 10-m depth at LTER1 on the north shore of Moorea (*Edmunds, 2018a*).

Rates of whole and partial colony mortality were estimated from previously published research that employed in situ tracking of *Acropora, Pocillopora* and *Porites* colonies, conducted annually between 2011 and 2013 at LTER1 (*Lenihan & Kayal, 2015*; *Kayal et al., 2018*; Table S1). Rates of whole colony mortality were based on the proportion of colonies that did not survive between sampling times, while rates of partial colony mortality were based on the proportion of colonies that experienced shrinkage between sampling times. Rates of whole and partial mortality for *Montipora* were based on measurements obtained for *Acropora*, as *Montipora* colonies were not included in the study by *Kayal et al. (2018)*. Linear growth rates for *Pocillopora* were based on results from *Evensen & Edmunds (2016)*, and linear extension rates for *Acropora, Montipora*, and *Porites* were obtained from published studies (Tables S1). Coral recruitment was measured using settlement tiles immersed for ~6 months and deployed sequentially from 2009 to 2016 on the North shore of Moorea (summarized in *Edmunds, 2021*). Tiles were immersed from January/February to August/September and from August/September to January/February. Recruitment was recorded by family, with results for Pocilloporidae and Poritidae used to parameterise *Pocillopora* and *Porites*, respectively. In turn, recruitment rates of

Acroporidae were split equally to parameterise recruitment for *Acropora* and *Montipora*, though these rates were later adjusted (detailed below).

The efficacy of the IBM was evaluated through a contrast of projected and empirical coral cover, and discrepancies between these values under control conditions (i.e., ambient $pCO_2$) were addressed by reviewing the parameter values established for vital rates. As coral recruitment in Moorea varied extensively among years (*Edmunds, 2021*), this vital rate was targeted for adjustment in order to reduce the discrepancy between empirical and projected coral cover. Recruitment rates were adjusted until empirical and projected cover of each genus after 7 years differed by <1%, and the revised recruitment rate then was used in the projections of coral cover under OA conditions.

## Quantifying the impact of OA on coral recovery following acute disturbances

The model was used to assess the impact of elevated $pCO_2$ on coral community recovery over 7 years following 2010, relative to coral community recovery under ambient conditions. The impacts of OA on coral community recovery were modelled in two steps. The first step implemented the effects of elevated $pCO_2$ on linear growth rates of corals in the absence of competition, with growth rates modified based on measurements of linear growth for each genus under elevated $pCO_2$. Based on experiments with *Pocillopora* and *Montipora* in Moorea, linear growth rates over 28 d were reduced by 30% (at 972 µatm $pCO_2$) and 43% (at 1,033 µatm $pCO_2$) relative to ambient conditions (*Evensen, Edmunds & Sakai, 2015*; *Evensen & Edmunds, 2016*). The linear growth rate of *Acropora* spp. under OA conditions has not been recorded in Moorea and, therefore, the effects were based on a study from Eilat, Israel, in which the linear growth of *Acropora* was depressed by 37% under elevated $pCO_2$ (1,795 µatm) compared to ambient conditions (400 µatm; *Horwitz, Hoogenboom & Fine, 2017*). Lastly, growth rates of massive *Porites* were considered to be unaffected by OA compared to ambient simulations in the present analysis, based on a study from Moorea in which elevated $pCO_2$ (804 µatm) had no effect on calcification rate (mass per area) of massive *Porites* compared to ambient $pCO_2$ (*Edmunds, Brown & Moriarty, 2012*). The second step implemented the effects of coral competition among corals on coral growth rates under elevated $pCO_2$, using the approach described above under Step 1. This two-step approach allowed the model to differentiate between simulating the effects of OA on colony growth alone and simulating the effects of OA on both colony growth and competition among corals. The advantage of this approach is that it supported separate consideration of two mechanisms by which elevated $pCO_2$ can impact coral communities.

For all scenarios, 100 simulations were conducted, from which the average coral cover was calculated to assess the impact of OA on coral recovery, relative to ambient $pCO_2$. All model simulations were conducted using MATLAB (v9.1 R2016b; MathWorks, Natick, Massachusetts, USA). Figures were plotted in R using 'ggplot2' (*Wickham, 2016*) or in Prism (v. 8.0; GraphPad Software, Inc., San Diego, CA, USA).

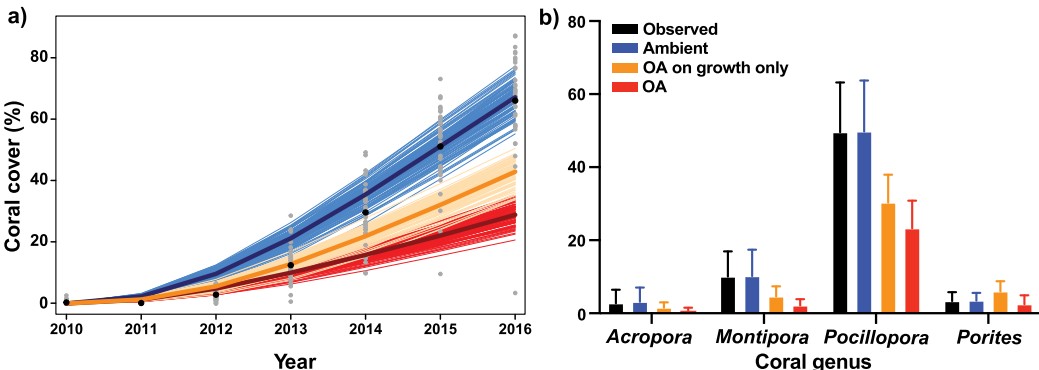

**Figure 3 Description of empirical and modeled coral community recovery at 10-m depth, at site LTER1 on the fore reef of Moorea, French Polynesia.** (A) Short-term observations and simulations (7 years) of total coral cover. Grey dots show observed coral cover in individual photoquadrats, with black dots depicting annual means. Thin lines show simulations of the recovery under ambient conditions (blue), with the effects of OA on coral growth alone (orange), and with the effects of OA on both coral growth and competition among corals (red), with the mean trajectory represented by the thicker line ($n = 100$). (B) Observed (black) and simulated cover of each coral genus after the 7-year recovery period. Bars are mean ± S.D. for observed data ($n = 65$) and 95% confidence intervals (CI) based on percentiles of 100 simulations.

## Sensitivity analysis

The relative impact of linear growth rates, competition among corals, recruitment, and partial and whole colony mortality on simulations of coral community recovery were assessed through a sensitivity analysis. Parameter sensitivity was evaluated based on the deviation of model projections with the ±20% change in each parameter after *Bozec et al. (2019)*, relative to the mean coral cover predicted after 7 years under ambient conditions.

## RESULTS

### Modelling the relationship between coral–coral competition and linear growth

For conspecific interactions under ambient $pCO_2$, the effects of competition on linear extension were best described by a linear relationship between growth reduction and percent contact (Fig. 2B), and for heterospecific interactions under ambient $pCO_2$, the effects of competition on linear extension were best described by an exponential decay model (Fig. 2C). Under elevated $pCO_2$, exponential decay models were better fitted to the relationships between linear extension and contact with conspecifics and heterospecifics under elevated $pCO_2$ (Figs. 2D, 2E).

### Model accuracy

Following 7 years of recovery initiated after coral cover was depressed to 0.3% in 2010, simulated coral cover reached 67.1% by 2016, a 1% difference from observed values, 66.1%, in 2016 (Fig. 3A). Recruitment rates of *Pocillopora* (4.15 corals m$^{-2}$) required a 10% increase to match observed rates of recovery, with *Pocillopora* cover reaching 50% after 7 years, compared to an observed value of 49.8% (Fig. 3B). In turn, *Montipora* cover reached 10.2% after 7 years, compared to an observed value of 10.1%, following a 142%

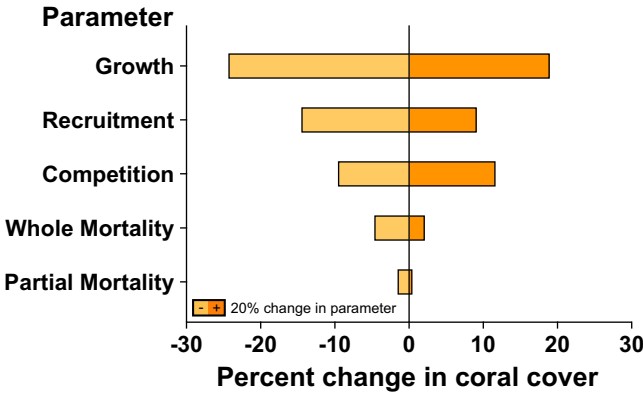

**Figure 4 Sensitivity analysis of model simulations to individual changes (±20%) in key parameter values for coral community recovery.** Effects of parameter changes are reflected by percent change from 67% coral cover after 7 years under ambient conditions.

increase in modelled recruitment rate (2.42 corals m$^{-2}$). *Acropora* recruitment (0.25 corals m$^{-2}$) was reduced by 50% in order for simulations to end with 3.2% cover, compared to the observed cover of 2.8% after 7 years. Lastly, *Porites* required the biggest adjustment in empirical recruitment rates in order for empirical and simulated cover of this genus to converge on ~3.5% after 7 years, with this outcome requiring a six-fold increase in empirical recruitment (3.9 corals m$^{-2}$).

## Rates of coral recovery under ambient and elevated pCO$_2$

When the effects of OA on coral growth were implemented, without the effects of OA on competition, coral cover following 7 years of recovery reached 42.9%, which was 24.2% less than projections of coral community cover under ambient pCO$_2$. In turn, when the effects of OA on both coral growth and coral–coral competition were implemented in the model, coral cover reached only 28.9% after 7 years (Fig. 3A), with *Pocillopora* accounting for 81% of the total coral cover at the end of the projections (Fig. 3B).

## Sensitivity analysis

The sensitivity analysis demonstrated that changes to coral linear growth rates had the strongest effects on model projection, followed by changes to recruitment rates and competition among corals (Fig. 4). A ±20% change in linear growth rate caused −24.4% declines and +19.1% increases in projected coral cover over 7 years; a ±20% change in recruitment led to −14.6% declines and +9.2% increases in projected coral cover over 7 years; and a ±20% change in competition led to −9.7% declines and +11.8% increases in projected coral cover. A ±20% changes to partial and whole colony mortality rate had little effect on the rate at which coral cover recovered following large-scale mortality, leading to −1.6% and −4.8% declines, and 0.6% and 2.2% increases, in projected coral cover over 7 years, respectively.

## DISCUSSION

The present study utilizes an IBM developed for a fore reef coral community to evaluate how the effects of OA (i.e., ~1,100–1,200 µatm $pCO_2$) on coral growth and competition among corals scale up to impact the projected rate of increase in coral cover following a major disturbance. Under current ambient $pCO_2$, the coral community at 10-m depth on the fore reef of Moorea has quickly recovered from recent disturbances, with coral cover rising from 0.3% in 2010 to 66% in 2016, primarily through high recruitment of *Pocillopora* (*Tsounis & Edmunds, 2016*; *Edmunds, 2018b*). However, the present simulations suggest that the rate of coral recovery over 7 years following a severe disturbance will be slowed by 56% under an OA regime of ~1,000–1,200 µatm $pCO_2$, relative to the recovery rate observed under present conditions from 2010 to 2016, with coral cover reaching 29% under OA after 7 years. Conversely, when the effects of OA were only applied to linear growth rates of corals, and not competition among corals, coral cover reached 43% after 7 years, 49% higher than the recovery when the effects of OA were implemented on both growth and competition among corals. The present analyses reveal the high potential for an IBM approach to improve understand of the extent to which coral cover will change in the future. Using a series of simplified assumptions and a limited set of empirical data to inform the model, the results indicate a strong effect of altered coral–coral competition under OA on coral community dynamics. Indeed, while elevated $pCO_2$ may not result in high rates of coral mortality, model projections suggest multiple pathways, including reduced colony growth and altered coral–coral competition, by which OA can slow coral community recovery following large-scale mortality events affecting reef-building corals.

There is a consensus that the growth rates of most scleractinians will be negatively impacted by OA (*Chan & Connolly, 2013*; *Kroeker et al., 2013*), yet these results, mostly from experiments in which colonies have been grown in isolation (i.e., without touching other colonies), have the potential to be modified by interactions among corals of the same or different species (*Evensen, Edmunds & Sakai, 2015*; *Evensen & Edmunds, 2016*; *Horwitz, Hoogenboom & Fine, 2017*). When the effects of OA on coral growth alone were included in the present projections, the rate of increase in coral cover was reduced by 39–54% for *Pocillopora*, *Montipora*, and *Acropora*, relative to present day $pCO_2$ conditions. Conversely, the cover of *Porites* increased by 70% after 7 years, as our model assumed the growth rate of this genus was unaffected by OA (*Edmunds, Brown & Moriarty, 2012*; *Comeau et al., 2014a*; *Edmunds & Yarid, 2017*). It is important to note that this assumption may be specific to Moorea, as studies conducted elsewhere have indicated a negative impact of OA on the growth of massive *Porites* (*Krief et al., 2010*). Nonetheless, the projected increase in cover of *Porites* in the present study likely was due to alleviated competitive interactions with the other coral genera. Natural $CO_2$ vents in Papua New Guinea have highlighted the potential for *Porites* to benefit from alleviated coral–coral competition under elevated $pCO_2$ conditions. Massive *Porites* increased in cover on reefs where $pCO_2$ reaches ~950 µatm and few other coral species were found, relative to nearby reefs experiencing ambient $pCO_2$ where coral cover and diversity were higher (*Fabricius et al., 2011*), and thus competitive interactions with other corals were more

likely to occur. Indeed, when the effects of OA for both coral growth and coral–coral competition were included in the present model, the cover of *Porites* decreased by 29% relative to present day $pCO_2$ conditions, with *Pocillopora* projected to persist on these reefs.

Including the effects of OA for both coral growth and coral–coral competition also resulted in further reduction in recovery for the other genera, ranging from a 53% reduction in cover after 7 years for *Pocillopora*, to a 79% decrease for *Montipora*, relative to present day conditions over the same period. Together, this led to a reduction in rate of increase in cover of the coral community to ~5% $y^{-1}$, compared to 11% $y^{-1}$ under ambient conditions. *Pocillopora* was the only genus to exceed 3% cover at the end of the projections under OA, reaching 23.4% cover. Indeed, although *Pocillopora* spp. has often been considered sensitive to acute disturbances (*Gleason, 1993*; *Darling et al., 2012*), *Pocillopora* spp. appears to be a diverse genus (i.e., consisting of at least six species including *P. verrucosa* and *P. meandrina* (*Edmunds et al., 2016b*; *Burgess et al., 2021*) that has overcome acute disturbances through high rates of recruitment in the 2–3 years following cessation of the COTs outbreak and cyclone Oli (*Bramanti & Edmunds, 2016*). High recruitment of *Pocillopora* on the fore reef of Moorea has resulted in a rate of increase in coral cover that exceeds that of any Indo-Pacific fore reef sites to date (*Graham, Nash & Kool, 2011*). Through larval connectivity, the projected persistence of this genus under OA could prove beneficial for other reefs in the region, because *Pocillopora* is ubiquitous across the Society Islands and throughout the tropical Eastern Pacific (*Magalon, Adjeroud & Veuille, 2005*; *Edmunds et al., 2016b*).

In contrast to *Pocillopora* and *Porites*, *Acropora* and *Montipora* showed limited potential to persist in this fore reef habitat under the OA regime tested, at least as evaluated from the low cover (<2.5%) established within 7 years of a major disturbance. *Acropora* and *Montipora* typically are sensitive to chronic and acute stressors (*Darling et al., 2012*), and yet these genera include some of the most ecologically important species on Indo-Pacific reefs, notably within *Acropora* (*Graham & Nash, 2013*). In the present study, it is possible that *Acropora* were underrepresented in the model projections, however, as the model did not account for the positive density-dependent recruitment observed for *Acropora* on the fore reef of Moorea (*Bramanti & Edmunds, 2016*). Positive density-dependent recruitment would favour recovery of *Acropora* populations as cover of this genus increases (*Kayal et al., 2018*). Still, the cover of *Acropora* and *Montipora* has remained low on the fore reef of Moorea since 2010 (*Adjeroud et al., 2018*), potentially resigning these genera to becoming ecological 'ghosts' (sensu *Hull, Darroch & Erwin, 2015*) at a local scale.

To implement an IBM to project coral cover in the present analysis, it was necessary to accept several limitations regarding the empirical data with which the model would be informed. Indeed, the necessity of adjusting recruitment rates to bring projected cover to within 1% of empirical cover highlights how little is known about the factors causing coral recruitment to greatly vary among years (*Edmunds, Leichter & Adjeroud, 2010*), or how this variation modulates changes in coral cover. Nevertheless, the adjustment to recruitment was highly effective for the dominant genus, *Pocillopora*. Additionally, there were uncertainties associated with quantifying the effects of OA on vital rates, while

consideration of the effects of OA on algal dynamics (e.g., on algal productivity, grazing dynamics, and coral-algal competition) was beyond the scope of the study. While macroalgae often can pre-empt space and limit coral recruitment on many reefs following disturbances (*Bender, Diaz-Pulido & Dove, 2012*; *Evensen et al., 2019*), macroalgae were not considered a major competitor for space in the present model, as there has been no evidence of macroalgal overgrowth on the fore reef of Moorea, even after the acute disturbances created by COTs and Cyclone Oli (*Adam et al., 2011*). Moorea may be a special case in this regard, because fish herbivory at 12-m depth on the fore reef in 2010 was sufficient to prevent macroalgal proliferation on reefs recently denuded of coral (*Holbrook et al., 2016*). While the lack of macroalgae in the model may still be representative of reefs where herbivores are abundant, the outcome of competition under OA may be different on reefs that lack adequate herbivory. Indeed, some macroalgae can benefit from increases in $CO_2$ and/or $HCO_3^-$ under OA (*Koch et al., 2013*; *Diaz-Pulido & Barrón, 2020*; *Ho et al., 2021*), enhancing the competitive ability of macroalgae versus corals, and possibly increasing macroalgal dominance on reefs under OA (*Anthony et al., 2011*; *Diaz-Pulido et al., 2011*; *Johnson et al., 2012*). As such, it is important for future studies to build on the present approach with more sophisticated models (i.e., that may include macroalgal dynamics where applicable) that can be developed as suitable empirical data become available.

Our model also did not consider the possibility for acclimatization or adaptation of corals to reduce their sensitivity to elevated $pCO_2$, which may have resulted in more pessimistic projections of coral cover. We also did not account for the potential of depressed coral recruitment under elevated $pCO_2$ (*Doropoulos et al., 2012*; *Fabricius et al., 2017*), which, in contrast to the possible positive effects of acclimatization and adaptation, may have favoured optimistic projections of coral cover. Indeed, sensitivity analysis suggest that coral recruitment can strongly affect projected coral cover, with a 20% decline in recruitment resulted in a ~15% decline in coral cover after 7 years. Lastly, the frequency and severity of bleaching events will almost certainly increase as ocean warming continues (*Heron et al., 2016*), further challenging the capacity of the coral community to maintain its resilience, possibly as has occurred following severe bleaching in 2019 (*Burgess et al., 2021*). Despite the aforementioned limitations of the present study, the outcome of this modelling effort provides an important advance towards understanding the impacts of OA on corals at a reef-scale. Clearly, more data on the effects of OA on coral vital rates and coral reef community processes are required, however, to better parameterise the models, effectively capture ecological realism, and improve the accuracy of model projections under climate change conditions.

## ACKNOWLEDGEMENTS

We would like to thank members of the CSUN Polyp lab who helped collect empirical data used to parameterize the model. We would also like to thank J.-C. Ortiz who helped adapt the model to include empirical data from Moorea, and members of the Marine Spatial Ecology Lab whose feedback helped shape this study. This is contribution no. 364 of the Marine Biology Program of California State University, Northridge.

### Funding

This work was funded by ARC and NESP grants to Peter J. Mumby, and funding from the National Science Foundation to the MCR-LTER (OCE 16-37396) and for OA research (OCE 14-15268) to Peter J. Edmunds. The funders had no role in study design, data collection and analysis, decision to publish, or preparation of the manuscript.

### Grant Disclosures

The following grant information was disclosed by the authors:
ARC and NESP.
National Science Foundation to the MCR-LTER: OCE 16-37396.
OA: OCE 14-15268.

### Competing Interests

The authors declare that they have no competing interests.

### Author Contributions

- Nicolas R. Evensen conceived and designed the experiments, performed the experiments, analyzed the data, prepared figures and/or tables, authored or reviewed drafts of the paper, and approved the final draft.
- Yves-Marie Bozec conceived and designed the experiments, performed the experiments, analyzed the data, authored or reviewed drafts of the paper, and approved the final draft.
- Peter J. Edmunds conceived and designed the experiments, performed the experiments, authored or reviewed drafts of the paper, and approved the final draft.
- Peter J. Mumby conceived and designed the experiments, performed the experiments, authored or reviewed drafts of the paper, and approved the final draft.

### Field Study Permissions

The following information was supplied relating to field study approvals (i.e., approving body and any reference numbers):

Permits for fieldwork were issued by the Haut-commissariat de la République en Polynésie Française (DRRT) (Protocole d'Accueil 2010–2011, 2011–2012, 2012–2013, 2013–2014, 2014–2015, 2015–2016 to PJE).

### Data Availability

The code of the simulation model is available at GitHub: https://github.com/ymbozec/REEFMOD.5.3.Ocean.Acidification

The model was parameterized using data from cited, published sources and from datasets that are available at:

Moorea Coral Reef LTER and P. Edmunds. 2020. MCR LTER: Coral Reef: Long-term Population and Community Dynamics: Corals, ongoing since 2005 ver 38.

Environmental Data Initiative. DOI 10.6073/pasta/10ee808a046cb63c0b8e3bc3c9799806 (accessed 2021-05-12).

Carpenter, R of Moorea Coral Reef LTER. 2020. MCR LTER: Coral Reef: Long-term Population and Community Dynamics: Benthic Algae and Other Community Components, ongoing since 2005. knb-lter-mcr.8.32 DOI 10.6073/pasta/0bf200e9e0f099de69826f57b18ff3da

Evensen, Nicolas R; Edmunds, Peter J (2016): Interactive effects of ocean acidification and neighboring corals on the growth of *Pocillopora verrucosa*. PANGAEA, DOI 10.1594/PANGAEA.867268, Supplement to: Evensen, NR; Edmunds, PJ (2016): Interactive effects of ocean acidification and neighboring corals on the growth of *Pocillopora verrucosa*. Marine Biology, 163(7):148 DOI 10.1007/s00227-016-2921-z.

## Supplemental Information

Supplemental information for this article can be found online at http://dx.doi.org/10.7717/peerj.11608#supplemental-information.

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
