# Peer review of "Scaling the effects of ocean acidification on coral growth and coral–coral competition on coral community recovery"

_PeerJ, doi:10.7717/peerj.11608_

## Round 0.1 · original submission · Major Revisions

· Academic Editor

Major Revisions

While the reviewers recognize the merit of this manuscript, they raised a number of issues to be addressed before further consideration of this ms for publication. Please send me your revised manuscript, together with a copy with tracked changes and detailed responses to comments of the reviewers (point-to-point format).

Reviewer 1 ·

Basic reporting

The study tests the possibility to apply model simulation and projection for coral recovery under ocean acidification, which is very meaningful in the future coral conservation. While I still have some concerns listed below.

Line 154-161, authors claimed to use Evensen & Edmunds (2016) results for model fitting. In the previous results, both vertical and horizontal growth were measured, so which growth rates are applied to the model in this study? Please be more specific.

Line 174-176, the competitive hierarchies were determined according to previous research, while the three researches were conducted in different regions (Taiwan, Israel and France). It’s reasonable that coral communities may maintain different competition hierarchies in different places. Although I think the hierarchy is acceptable for the present model, more evidence is needed if you’re trying to clarify there is consistency in coral competition across the regions or please find researches conducted in similar regions to support this statement.

Fig.2(a) The experiments were conducted and published in previous study and not the main purpose of the present one. It’d be better to move it to supplementary material.

Experimental design

Line 180-182 I can easily understand the meaning of equation 1. While for equation 2 in Table 1, where does it come from? Do you have any reference or just build it from your data trend?

Line 248-251, all the analysis code, including this model simulation, which is a major part of the study, should be provided in the supplementary material.

Validity of the findings

Line 328: The model assumed the growth rate of Porites was not affected by OA, this assumption is not justified. There are evidences that show a significant decline in growth and calcification of Porites under OA.

Line 377-382: Macroalgae is the main competitor in most disturbed reef, thus exclude this from the model could prevent its general application.

Reviewer 2 ·

Basic reporting

no comment

Experimental design

no comment

Validity of the findings

no comment

Additional comments

Evensen et al addressed a very important while largely neglected topic in this area and combined empirical data and modelling approach to scale up the OA effects on the dynamics and recovery of coral communities. They found that when considering coral-coral competition in the model, the effect of OA on coral recovery was stronger. Generally, this is a timely and excellent study. The writing and logic flow is very good for easy reading. I only have one major concern and several specific comments
Although the authors did state that Macroalgae was not considered a major competitor for space in the present model and algal cover remain <5% in Moorea. According to figure 1, many fleshy macroalgae adjacent to corals can be clearly seen from the quadrat. How much do the authors think that this may influence your modelling approach and results? I think that it is better to relook at all the photoquadrats and see how many corals are surrounded by fleshy algae?
Specific comments:
Line 130-131 “colonies of all four genera were assumed to extend linearly in planar view by 1.5 cm every 6 months, representing their average growth over 6 months (Table S1).” After checking table S1 and calculation codes, I still cannot get how you can get the variable linear growth rate of all four genera into a uniform the planar extension rate of 1.5cm? This is very important but it was not clearly explained as the explanation about the coral-coral competition in figure 1.

Line 133 photoquadrat: What is the size of quadrat and are they equal to the cells in your modelling? are these quadrats fixed or not between 2010 and 2016?

Line 371 remove “the” before how little
Figure 2 caption: relationship between planar growth (as a proportion of the maximum growth rate). If all data are expressed in proportion to the max, they should be below 1 and I noticed that there are many datapoints beyond 1.

---

## Round 0.2 · accepted · Accept

· Academic Editor

Accept

The reviewers considered that the authors have adequately addressed the issues raised and recommended this manuscript for publication.

Reviewer 1 ·

Basic reporting

My concerns have all been well addressed. I have no further comment.

Experimental design

No comment.

Validity of the findings

No comment.